# LMVSegRNN and Poseidon3D: Addressing Challenging Teeth Segmentation Cases in 3D Dental Surface Orthodontic Scans

**DOI:** 10.3390/bioengineering11101014

**Published:** 2024-10-11

**Authors:** Tibor Kubík, Michal Španěl

**Affiliations:** 1Department of Computer Graphics and Multimedia, Brno University of Technology, Božetěchova 2, 612 66 Brno, Czech Republic; spanel@fit.vut.cz; 2TESCAN 3DIM, s.r.o., Libušina tř./21a, 623 00 Brno, Czech Republic

**Keywords:** dental scans, tooth segmentation, 3D mesh segmentation, Poseidon3D, Poseidon’s Teeth 3D, LMVSegRNN, orthodontic mesh segmentation dataset

## Abstract

The segmentation of teeth in 3D dental scans is difficult due to variations in teeth shapes, misalignments, occlusions, or the present dental appliances. Existing methods consistently adhere to geometric representations, omitting the perceptual aspects of the inputs. In addition, current works often lack evaluation on anatomically complex cases due to the unavailability of such datasets. We present a projection-based approach towards accurate teeth segmentation that operates in a detect-and-segment manner locally on each tooth in a multi-view fashion. Information is spatially correlated via recurrent units. We show that a projection-based framework can precisely segment teeth in cases with anatomical anomalies with negligible information loss. It outperforms point-based, edge-based, and Graph Cut-based geometric approaches, achieving an average weighted IoU score of 0.97122±0.038 and a Hausdorff distance at 95 percentile of 0.49012±0.571 mm. We also release Poseidon’s Teeth 3D (Poseidon3D), a novel dataset of real orthodontic cases with various dental anomalies like teeth crowding and missing teeth.

## 1. Introduction

Surface dental scans are increasingly popular in digital computer-aided dentistry intervention planning [1]. One of the fields where teeth surface scans are widely used is digital orthodontics and 3D printing of clear aligners [2]. For diagnosing and preparing the patient’s treatment plan, individual teeth must be precisely recognized and subsequently rearranged into a normal occlusion. Manual or error-prone (semi-)automatic segmentation of orthodontic cases requires clinicians to undertake laborious and time-consuming actions until satisfying results are produced.

Automatic and robust teeth segmentation is non-trivial for at least a couple of reasons. One difficulty stems from the high prevalence of dental anomalies and irregularities, particularly among orthodontic patients. The patient’s arch might be asymmetric or incomplete (caused, for example, by extractions or retention). Positional tooth anomalies are also frequently seen among patients (rotations, teeth transpositions, or transmigrations). Occasionally, patients wear dental appliances such as braces or wires during dental scan acquisition. Finally, depending on the clinician’s proficiency, the resulting mesh might contain holes, blurred geometry, incomplete scans (completely omitted or insufficiently scanned molars), and others. Second, unlike the volumetric representation obtained from CT or CBCT scanners, surface scanning devices capture optical impressions as 3D shapes. The direct application of operations like convolution and pooling is, hence, more challenging.

An automatic solution capable of segmenting teeth in 3D surface scans robustly in all mentioned aspects is indispensable to streamlining clinicians’ workflow and focusing more attention and focus on treatment planning. The aim of this study is to provide a public dataset of such cases and also a method that handles the segmentation of these cases with the minimum necessary adjustments.

### 1.1. Related Works

A considerable number of frameworks aim at addressing the tooth segmentation task in 3D surface scans [3,4,5,6,7]. Research in this line mainly utilizes point cloud representation (point based), frequently combined with graph learning techniques (graph based). Instead of relying on PointNet [8] or PointNet++ [9], which are agnostic to capturing local geometric contexts, researchers often introduce more complex and sophisticated methods. One example is MeshSegNet [3], where the authors propose a dense fusion strategy to combine local-to-global geometric features for mesh cell labeling. Although it provides promising results in simple cases, its performance in real-world orthodontic scenarios remains uncertain since the evaluation was conducted on a dataset with restricted variability and complexity. Furthermore, the method relies on a Graph Cut-based refinement process sensitive to parameter tuning. Another point-based method, TSegNet [4], follows a detect-and-segment approach, enabling the localized segmentation of teeth masks. Its limitation lies in the fact that the detection module identifies tooth centroids, which are not very intuitive to adjust by clinicians. Consequently, any inaccuracies in the detection step leave no room for intermediate corrections and can lead to substantial errors in the segmented teeth masks. This leads to more labor-intensive refinements of the segmentation masks. The works discussed in references [6,7] represent notable efforts towards clinically applicable solutions that do not rely on a large-scale annotated dataset. They achieve this by introducing self-supervision techniques and training with weak annotations. These detect-and-segment solutions demonstrate the feasibility of utilizing highly accurate detectors to significantly enhance the performance of segmentation networks. As aforementioned papers, they are homogeneous in terms of point cloud utilization.

Two projection-based approaches have emerged in the literature [10,11]. The method referenced in [10] encodes dental mesh into a plan view and panoramic depth image. It relies on hand-crafted feature engineering rather than employing a deep learning framework. In [11], the authors propose a deep learning approach, where each mesh face is encoded into a feature vector of geometric attributes. Each such descriptor is converted into a 2D feature map, an input to a 2D CNN. Since each encoded face is processed separately and the method relies on optimization algorithms, it is computationally heavy, which may present challenges in practical real-time applications. Additionally, the spatial relationships within these maps can be challenging to exploit by 2D convolutions, as the values are not inherently spatially correlated. Projection-based frameworks benefit from the well-explored field of 2D CNNs and are usually very efficient in training and inference speed. To widen the research field with a well-designed projection method, it begs the following question: could a projection-based method be on par with or even outperform 3D geometric approaches in robust mesh teeth segmentation?

A common characteristic among all the teeth segmentation frameworks discussed is the unavailability of publicly released data. Consequently, any attempt to compare these methods lacks complete fairness, making it unclear which approach stands out as superior. Another common attribute is the heavy downsampling of the input point cloud, losing important anatomical features captured in the fine geometric details.

In this work, we propose a projection-based method that does not need any mesh decimation and effectively builds upon the detect-and-segment idea. Opposite to previous projection-based approaches, we represent the 3D shape as a sequence of 2D maps that locally encode the depth and geometry of each tooth in a form, where functional proximity coincides with physical proximity, which introduces an important inductive bias to 2D learning. To the authors’ best knowledge, there are no projection-based methods employing CNNs for the segmentation of teeth in 3D mesh data represented as multiple geometry maps. The mesh segmentation framework MV-RNN [12] is methodologically closest to our work. It builds upon MV-CNN [13], where the shape is rendered and evaluated from various viewpoints, with the final joint optimization into a compact object descriptor. MV-RNN extends MV-CNN by incorporating recurrent blocks that correlate predictions across different views. While our teeth segmentation framework draws inspiration from MV-RNN, it diverges in several key aspects: (i) we apply our method to segment anatomical shapes characterized by intricate geometry and significant shape variability rather than low-frequency shapes representing human bodies; (ii) our method is specifically tailored for the tooth segmentation task, involving a localized analysis of each tooth to enhance the geometric information at the network’s input; (iii) we utilize Bi-Directional ConvLSTMs [14], which preserve more spatial information as opposed to LSTM [15]; and (iv) instead of relying on shaded images, we render input maps with rich geometric information (depth and curvature) invariant to changes in lighting, shading, and reflectance.

We believe in the potential of this projection-based approach, based on our previous work on the tooth detection task on digital dental casts [16] as well as on our experiments introduced in this paper. Our results show that networks can effectively learn features for the perceptual parsing of 3D medical shapes from rendered images. Our work presents a methodologically different method from the usual practices in neural tooth segmentation, which is valuable for broadening the understanding of the problem.

### 1.2. Contributions

Our contributions to the field can be summarized as follows:Streamlined segmentation. Our method operates in a detect-and-segment manner, where detected surface landmarks (rather than tooth centroids) offer practical advantages for clinicians, enabling quick corrections compared to laborious adjustments of segmented 3D regions. This demonstrates significant potential to reduce human efforts in orthodontic treatment planning.Information-rich projections. Utilizing predictions from the detector, our segmentator addresses typical projection issues related to unclear camera positioning and unknown shape orientation. It allows for localized segmentation and reduces the task to a binary segmentation problem. Given the general convex ovoid tooth shape and prior landmark detection, the multi-view configuration can be preset, so pose sensitivity is not an issue. Alongside the local tooth processing, we show that the rendered maps capture the 3D shape with minimal information loss.Highly competitive performance. We achieve an average weighted IoU score of 0.97122±0.038 and a Hausdorff distance at the 95th percentile of 0.49012±0.571 mm. We provide a comparative analysis with other point-based [8,9] and mesh-based [17] methods and with a Graph Cut-based solution. Our method outperforms each of the methods.Release of Poseidon3D dataset. We address a notable gap in the field by introducing a new dataset of real-world orthodontic cases of increased anatomical complexity. We name it Poseidon’s Teeth 3D (Poseidon3D). This dataset contains 3D scans of patients who exhibit various dental anomalies like teeth crowding, damaged and missing teeth, and teeth with visible roots. To the best of our knowledge, we are the pioneering contributors of such a dataset, and we anticipate that it will establish a standardized basis for fair evaluation in future studies.

## 2. Poseidon3D: Dataset of Anatomically Complex Dental Shapes

A new public dataset for tooth segmentation in 3D surface scans is presented. It is named Poseidon’s Teeth 3D (Poseidon3D). Poseidon3D contains 200 3D triangular meshes: 88 maxillas and 122 mandibles.

All data samples are real orthodontic cases scanned by various intra-oral scanners and by indirect scanners of wax impressions. The scanning process was handled by a professional clinician prior to the orthodontic procedure. Example cases are shown in Figure 1. All cases were anonymized, so it was not feasible to undertake a thorough analysis concerning ethnicity, gender, or age. However, the data come from multiple clinical sites across several continents. Shapes in Poseidon3D retain geometry information on vertex positions and topology, with an average face count of approximately 200,000, and do not carry any information on colors or textures.

Individual tooth regions for each case were accurately labeled by a clinician and subsequently reviewed and refined to ensure the high quality of the labels. A single connected component always forms each region.

Poseidon3D was collected by focusing on patient cases that noticeably deviated from healthy full dental arches. Cases containing teeth with significant orthodontic anomalies were accumulated, such as diastema (abundant space between upper incisors), abundant space between teeth in general, infraocclusion and retention (partially erupted teeth that do not reach the occlusal plane), mesial and distal inclination, multiple types of crowding of various severity, missing teeth, damaged teeth, teeth with visible roots, present third molars, and others.

Compared to the only publicly available dataset called *3DTeethSeg22* [18], the presented dataset is smaller in size (200 vs. 1800 mesh models). Still, it addresses significantly more abnormalities naturally present in orthodontic procedures. We anticipate that Poseidon3D will provide insights into the generalization capabilities of novel methods, particularly in their practical application in routine clinical practice, where abnormalities in dentition are prevalent. We randomly select five representative meshes from Poseidon3D and 3DTeethSeg22 and provide the visual comparison in Figure 2. Twenty simple cases are included for additional evaluation of the generalization to complex cases.

All data samples in STL format, along with per-face annotations of teeth masks, are available at https://tiborkubik.github.io/lmvseg-poseidon3d/ (accessed on 10 October 2024).

## 3. Method: High-Level Overview

A schematic view of the proposed framework is given in Figure 3. Given a set of vertices, $V\subseteq R^{N\times 3}$, a set of edges $E\subseteq V^{2}$, and a set of faces F={(k,l,m)∣k,l,m∈V∧(k,l),(l,m),(m,k)∈E}, input mesh is defined as M=(V,E,F). There are no assumptions about M concerning manifoldness, closeness, resolution, or orientation. It is expected to be a geometric object representing a single human dentition (either a maxilla or mandible). The objective of our study is to model the function g⊆F×C, where C is a set of classes such that C={L1,L2,…,L8,R1,R2,…,R8,G} and |C|=17, representing sixteen classes for individual teeth regions encoded as *L*(eft) or *R*(ight) quadrant followed by a number starting from the central incisor to the third molar (one to eight), and one class *G* that represents gingiva. Intuitively, function *g* assigns each face from M a specific class representing 1 of 16 teeth or gingiva.

Note that mapping *g* is not surjective since, in most cases, the scanned dental arch is incomplete, i.e., teeth are missing. To model *g*, we employ a detect-and-segment approach.

### 3.1. Detector

The proposed method runs the automatic tooth detection presented in [16]. It detects two occlusal/incisal landmarks on the surface of each tooth by running a multi-view 2D pipeline coupled with RANSAC-based post-processing for outlier removal. Such landmark positions (in contrast to tooth centroids) offer quick adjustability, allowing clinicians to review intermediate results and make efficient corrections as needed. The method is also robust to missing teeth in dental arches. The detector predicts vector $T_{toothPos}=\left[\left(m_{L8},d_{L8}\right),\left(m_{L7},d_{L7}\right),\ldots ,\left(m_{R8},d_{R8}\right)\right]$; $|T_{toothPos}|=16$, where each tuple $\left(m_{\left[LR][1-8\right]},d_{\left[LR][1-8\right]}\right)$ is formed by two vectors in $R^{3}$, representing the position of the distal and mesial landmark on the corresponding tooth surface, respectively. If a tooth is missing, the corresponding positions in $T_{toothPos}$ are set to infinity. In practice, the clinician would see the intermediate results of the detector and correct any inaccuracies, which is more time effective compared to the tuning of segmentation masks.

### 3.2. Occlusion Alignment

Positions from $T_{toothPos}$ (with excluded infinity values) are used, as well as a set of reference landmark positions, to align the 3D shape into a canonical position. The ICP algorithm [19] is run on these two sets of points in $R^{3}$ to obtain a rigid transformation matrix $M_{ICP}=\left[R|t\right]$, where R∈SO(3) (3D rotation group in geometry) denotes rotation and $t\in R^{3}$ denotes translation. $M_{ICP}$ is applied to M to align the scan to be upright oriented along the z-axis and centered around the origin. Note that within the new coordinate system, the approach makes no distinction between the maxillas and mandibles, handling both types of jaws in a unified way.

### 3.3. Segmentator

Within the coordinate system of occlusion-aligned meshes, we run the segmentation of individual teeth. Since the segmentator is the main body of the work, its detailed description is presented in Section 4.

## 4. Method: Segmentator

Teeth are segmented in two stages: (1) per-tooth stage, which is applied on each detected tooth separately, and (2) scan-level stage, where per-tooth predictions are merged and post-processing corrections are applied. The segmentator uses localization and identification information from the detector stage and analyzes 3D shapes transformed in canonical position by applying $M_{ICP}$.

### 4.1. Per-Tooth Stage

Individual teeth regions are obtained in a multi-view manner. Unlike common multi-view methods, our segmentation method overcomes concerns related to information loss and the ambiguity of camera extrinsic setup. First, leveraging prior tooth detection, each tooth is independently analyzed within its local neighborhood. This approach captures rich geometric information within a 2D image of compact resolution. Second, operating within a coordinate system featuring canonically aligned meshes and considering tooth crowns typically exhibit a convex ovoid shape when viewed from the occlusal surface, the extrinsic of the virtual cameras in a multi-view configuration can be accurately predetermined. So, for each tooth with corresponding values $T_{toothPos}$ not in infinity, the following subsections describe the sequence of steps executed for each such tooth.

#### 4.1.1. Input Map Generation by Next-View Sampler

The initial camera position is determined based on $T_{toothPos}$ of the currently processed tooth, calculated as the mean of the corresponding landmark positions. An additional offset of 15 mm is added to the z-position of the camera. This setup allows for the comprehensive capture of the entire occlusal surface of the tooth, including the surrounding neighborhood, thereby preserving the contextual information of adjacent teeth. Subsequent views are obtained by systematically moving the camera in a spiral-like manner around the tooth, maintaining a constant angular increase. Throughout this process, the camera consistently points toward the mean position of the tooth landmarks. An orthographic camera is used to obtain the input maps.

#### 4.1.2. Extraction of 2D Segmentation Masks

From each camera viewpoint *v*, two input maps of size H×W are generated via orthogonal projection: $D_{v}\in {\left[0,\;1\right]}^{H\times W}$, the depth map where 0 indicates infinity, and $N_{v}\in {\left[-1,\;1\right]}^{3\times H\times W}$, the normal map with the normal vectors in screen space. These maps make the method robust against a wide range of challenges, including variations in lightning, shading, and reflectance models. It retains essential geometric details, accurately capturing depth and curvature information. The input to the network can be written as $I_{v}:=\left\{D_{v},\;N_{v}\right\}\in R^{4\times H\times W}$.

To follow the multi-view approach, there is a set of such inputs $I=\{I_{v,1},\ldots ,I_{v,n}\}$, where $I_{v,i}$ is the map from the *i*-th (1≤i≤n) position of camera, where *n* is the total number of viewpoints.

An additional scalar signal is derived from the tooth type notation ([*LR*][1–8]). Each distinct tooth type is associated with a unique floating-point value $i_{tt}\in \left[0,\;1\right]$. These values lie in the unit range to align with the overall distribution of *I*. Including $i_{tt}$ is a significant inductive bias that allows for efficiently sharing a substantial portion of the network’s capacity for shared feature extraction. Then, the signal activates specific weights trained to fine-tune the results for the given tooth class, enhancing the model’s tooth-specific performance.

Each element of *I*, accompanied by the signal $i_{tt}$, serves as input for the segmentation network referred to as ${\mathrm{CNN}}_{1}$, adopting the U-Net [20] shape. As outputs of ${\mathrm{CNN}}_{1}$ have no inter-viewpoint context, they require further tuning by ConvLSTM to ensure the correlation of views and the consistency of the boundaries of the segmented regions. This input sequence *I* can be generated before the 2D segmentation mask prediction, allowing for the use of Bi-Directional ConvLSTM [14]. The output for each tooth is then a sequence of 2D segmentation masks $O=\{O_{v,1},\ldots ,O_{v,n}\}$, where each $O_{v,i}\in {\left\{0,1\right\}}^{H\times W}$. It indicates that we define the task as binary segmentation. Class information is incorporated into the network through the tooth signal $i_{tt}$. We argue that the binary segmentation task is easier to learn for the neural network.

#### 4.1.3. Unprojection of the Predictions to 3D

In this step, the sequence of output maps *O* is transformed into a set of faces $F_{C}\subseteq F$ representing the segmented tooth. For each output map $O_{v,i}$, rays are cast orthogonally only through those pixels of $O_{v,i}$, whose value is 1. For each face f∈F hit by any of the rays, we then set $F_{C}$ to $F_{C}\cup \left\{f\right\}$.

### 4.2. Scan-Level Stage

Upon completing the per-tooth stage, a post-processing step is carried out on the entire model. Two major issues might arise from the independent generation of tooth region masks: (1) some faces f∈F may be assigned to multiple classes from C, and (2) the unprojected regions do not form a single connected component.

#### 4.2.1. Region Voting

Our custom proposed region voting solves the situation when faces f∈F are assigned to more then one region. To tackle this, we define *multi-view certainty*, a relation u⊆F×C×P, where P={p∣p∈[0,1]}. Each tuple $(f_{i},c_{i},p_{i})\in u$ defines the certainty $p_{i}$ of face $f_{i}$ being a member of class $C_{i}$, and is obtained as the ratio of the number of views from which the face was hit by a cast ray representing the class $C_{i}$ and the number of views from which the face was initially visible.

For clearer demonstration, the relation can be perceived as mapping u:F×C↦[0,1]:(1)$$F\leftarrow \left\{\begin{array}{cc} \arg\max_{c\in C}\;u\left(f,c\right) & \mathrm{if}\;\max_{c\in C}\;u\left(f,c\right)\neq 0,\\ G & \mathrm{if}\;\max_{c\in C}\;u\left(f,c\right)=0. \end{array}\right.$$

This means that each face is assigned to the class with the highest multi-view certainty or to the gingiva class if the certainty is 0 for all classes.

#### 4.2.2. Connected Component Analysis

We assume that a single region forms each tooth in a dental scan. Therefore, any outlying faces are removed. The CCA approach [21] from graph theory is utilized. In case there is a class that is represented by more than one component, only the component with the highest face count is preserved. Faces of the remaining components are associated with classes respecting the multi-view certainty.

### 4.3. Implementation Details

${\mathrm{CNN}}_{1}$ is designed as a U-Net [20] with the following hidden features: [32,64,128,256,512] (five stages), and additional batch normalizations added in each stage to reduce overfitting between the convolutional and ReLU layers. Transposed convolutional layers accomplish upsampling. The recurrent unit (ConvLSTM) has 32 hidden states in both layers. The method is trained on sequences of 49 maps. Rendered maps have a spatial resolution of 256×256. This setting gives the best results concerning computational speed and memory requirements.

## 5. Experimental Results

The experiments discussed in this work were run on a system with 24 GB NVIDIA GeForce RTX 4090 and a 12-core AMD Ryzen 9 7900 with 64 GB RAM. Training of the best-performing model took approximately 2 days. The inference time depends on the number of viewpoints employed and falls within the range of seconds.

### 5.1. Training and Data Augmentation

Dental scans of the Poseidon3D dataset are divided into two groups: those used in the training procedure and those used for the evaluation. From the 200 3D shapes, 160 models are used for the network training, and 40 models are used to evaluate the proposed method. Furthermore, 160 meshes of training phase are split in the ratio of 4:1 into a training set and validation set, respectively.

To improve the generalization ability, we augment the training and validation data samples. We apply different transformations on the 3D shapes to generate augmented meshes. For each original surface already pre-aligned to occlusion via $M_{ICP}$, we generate 10 random rigid transformations, each represented by a matrix $M_{AUG}=\left[R|t|S\right]$, where R∈SO(3), $t\in R^{3}$ and *S* denote the scaling operation. We rotate the meshes with a random angle uniformly sampled within $\left[-\frac{\pi }{4},\frac{\pi }{4}\right]$ around an arbitrary axis. The vector *t* displaces the mesh by translating it by a value uniformly sampled between [−5,5]. The dental scan is resized by an isotropic scaling value uniformly sampled between [0.8,1.2]. Then, the sequence is generated without additional augmentations on the 2D maps. No augmentation is applied in the testing phase.

The network is optimized via batches of size 4, using AdamW optimizer with a learning rate and a weight decay of 0.001. Cosine annealing restarts are applied every 10,000 iterations. Models are trained by minimizing the Dice loss $L_{Dice}$ [22]:(2)$$L_{Dice}=1-2\frac{\sum_{n=1}^{N}p_{n}r_{n}+\epsilon }{\sum_{n=1}^{N}p_{n}+r_{n}+\epsilon },$$
where $r_{n}$ are values from reference foreground segmentation image of *N* pixels, $p_{n}$ are image elements of the predicted probabilistic map for the foreground labels, and ϵ is a term to ensure the loss function stability.

To provide shorter paths for gradients to flow during backpropagation, deep supervision [23] is employed, so the final loss value is computed as follows:(3)$$L_{Dice}^{\;total}=\sum_{i}^{S-1}L_{Dice}^{\;i}+L_{Dice}^{\;final},$$
where $L_{Dice}^{\;i}$ is the loss value computed at stage *i* of *S*-stage decoder and $L_{Dice}^{\;final}$ corresponds to the loss value computed from the output layer.

For the detection module, we follow the setup as presented in [16].

### 5.2. Evaluation Metrics

To evaluate individual methods and setups, two complementary metrics are employed that are suitable for the segmentation task [24]. Although the method operates in 2D projections, all metrics are measured on the final unprojected segmentation masks in 3D.

Let $\hat{V}$ be a point cloud of ground-truth boundary vertices, and V be a point cloud of prediction boundary vertices. Let *d* be the Euclidean distance in $R^{3}$. We define the Hausdorff distance $d_{H}$ as our boundary metric, and it is calculated as
(4)$$d_{H}\left(\hat{V},V\right)=\max\left\{\max_{x\in \hat{V}}\min_{y\in V}d\left(x,y\right),\max_{y\in V}\min_{x\in \hat{V}}d\left(x,y\right)\right\}.$$

$d_{H95}$ is its 95th percentile. This boundary metric detects artifacts in segmentated masks, such as missing details or narrow protruding parts.

We choose Weighted Intersection over Union as the overlap metric. In contrast to the standard IoU computed over meshes, W-IoU takes into account that different triangle sizes contribute differently to the final value:(5)$$W-\mathrm{IoU}=\frac{\sum_{f\in F_{\cap }}A_{f}}{\sum_{f\in F_{\cup }}A_{f}},$$
where $A_{f}$ is area of face *f*, $\hat{F}$ and F are faces of ground-truth and prediction masks, $F_{\cap }=\hat{F}\cap F$, and $F_{\cup }=\hat{F}\cup F$. *W-IoU* detects errors such as under/oversegmentation, or shift errors. We consider an overlap metric that does not weight values by the polygon area inaccurate.

### 5.3. Competing Methods

Using the Poseidon3D dataset introduced in Section 2, we compare the performance of our projection-based method with two representatives of point-based approaches (PointNet [8], PointNet++ [9]), one representative of edge-based approaches (SparseMeshCNN [17]) and one representative of the traditional segmentation algorithm based on the Graph Cut method. Their brief summarization follows.

Localized PointNet [8]: We implemented the architecture by following the original paper. The input is N×3 or N×6, where each row denotes the 3D coordinates of a mesh cell, possibly extended by its normal vector. To make the comparison more fair, we cut a submesh of the original dental scan for each tooth, following the localized processing. We sample N=8192 points from the mesh cut by farthest point sampling (FPS). We train the network with a batch size of 8 and point Dice loss.Localized PointNet++ [9]: This method intuitively extends PointNet by hierarchical modeling of spatial relations between neighboring points. Our architecture follows the segmentation architecture presented in the original paper. We apply the same point sampling strategy as in the case of PointNet. Similarly to PointNet, each tooth is analyzed locally, and the task is formulated as binary segmentation. Sampling and training configuration also remain the same.Localized SparseMeshCNN [17]: This edge-based method directly taps into the geometric nature of meshes and redefines convolution over mesh edges. To build hierarchical networks, mesh pooling via edge collapse is used. SparseMeshCNN extends the original MeshCNN by using sparse tensors. This network is again used to predict binary masks on mesh cuts in a detect-and-segment manner. As in the original work, the input feature for every edge is a 5d vector: the dihedral angle, two inner angles, and two edge-length ratios for each face. Each mesh cut is decimated to contain approximately 8000 edges.Graph Cut: This is a conventional segmentation method based on classic 2D image Graph Cut [25]. In 3D shape analysis, the graph is constructed from the mesh structure, with edge weights reflecting local curvature information. The Graph Cut algorithm is then applied to find the optimal cut, separating teeth from gum while following regions of lower curvature [26].

### 5.4. Results

We present the results along two experimental axes: an ablation study demonstrating the impact of key method components and a comparative analysis to show the competitive performance of a projection-based approach compared to geometric methods.

#### 5.4.1. Components of LMVSegRNN Gradually Improve Its Performance

Quantitative results of the ablation study are summarized in Table 1. The results regarding the input format reveal a couple of key insights.

Input format: First, depth information is crucial for robustly representing intricate tooth shapes, while normal maps alone represent the teeth less informatively. Second, stacking normal maps with depth maps enables learning a promising inductive bias for improving the results at the tooth–gingiva boundary. Curvature information in normal maps enables the network to generate smoother and more consistent segmentation results, particularly in complex and irregular boundary regions.Tooth type input signal: Incorporating additional input signals that convey tooth type information from the decoder delivers better results, speeds up training convergence, and enhances training stability. These findings support the hypothesis that adopting a detect-and-segment approach is beneficial.Deep supervision: Auxiliary segmentation heads in the intermediate layers of the decoder, optimized with the same loss, increase training stability and speed up convergence. These aspects are essential in overfitting prevention when learning from medical data.Recurrent units: Without imposing an inter-view context, the generated segmentation masks exhibit inconsistent region boundaries, appearing unnatural in some cases, featuring spiky and jagged artifacts. The addition of recurrent units improves the smoothness of region boundaries, which is particularly important for practical applications, as it reduces the need for minor and potentially unnecessary adjustments that do not affect the method’s usability. The performance gain (approximately 2% in overlap metric for both simple and complex cases) also suggests that including LSTM layers enhances the understanding of the 3D nature of the data.Region voting: Introducing the region voting procedure results in substantial improvement (2% on complex test set). Each tooth region is assessed independently and is successively back-projected to the mesh surface sequentially. Any inaccurate segmentation (typically predicted from one viewpoint) is corrected thanks to the computed multi-view certainty. The substantial impact of this module can be seen in Figure 4. It is evident that it suppresses significant errors on surfaces close to areas of contact with neighboring teeth. It allows us to define a multi-class task as single-class segmentation without any faces assigned to multiple classes.CCA: The impact of CCA on performance is minor when considering overlap metrics but notably impactful when considering boundary metrics (almost 0.4 mm improvement in $d_{H95}$ on complex cases). This finding is a notable sign of why combining overlap and boundary metrics in segmentation tasks makes sense.

#### 5.4.2. LMVSegRNN Outperforms Conventional Geometric Approaches

Our proposed method outperforms the Graph Cut-based framework, specifically tailored for tooth mesh segmentation. As evident in Table 2, the performance is almost on par with our solution (1.5% worse on *W-IoU* metric then our performance). However, in complex cases, the performance is worse by 2.6% on *W-IoU*. This suggests the effectiveness of our learning-based method, which better generalizes to atypical cases compared to the rule-based algorithm, even though explicitly tweaked for dental mesh segmentation.

#### 5.4.3. LMVSegRNN Outperforms Learning-Based Geometric Approaches

The qualitative results obtained by the competing methods are summarized in Table 2. Our projection-based method consistently surpasses the performance of other approaches based on representation learning. More importantly, it shows the greatest generalization ability to complex atypical cases and, thus, the highest potential to be exploited in daily clinical practice.

In competing point-based methods, including curvature information in the input representation is beneficial, which confirms and ties well with the results of our ablation study. The complete absence of local information in PointNet solutions and the very limited local information in PointNet++ methods, however, result in an inferior ability to uncover intricate patterns in the geometry of dental shapes, making the methods less accurate. In contrast, our projection-based method is built on hierarchical processing of depth and normal information via local 2D convolutions with 3D information shared among different views through recurrent units, effectively processing the input scan with information at different scales. Although much information may be lost by reducing to 2D, we argue that considerably more information is lost by strong undersampling of inputs required by point-based methods.

The results of the qualitative analysis support the findings presented above. Refer to Figure 5 for segmentation results produced on two randomly selected test cases. By visually examining the performance of individual approaches, we deduce several valuable insights regarding their typical error patterns. Outputs for the PointNet-based approach without information about normal vectors are characterized by oversegmentation. As there is no curvature information, the method learns blobs around teeth that are not bounded by the tooth–gingiva transition. On the other hand, the remaining point-based methods produce under-segmented masks. During the qualitative evaluation of edge-based methods, we observe that the generated masks occasionally exhibit holes. We argue that the results of point- and edge-based methods are not practically applicable due to the considerable effort needed to tune most results. The Graph Cut solution provides more applicable results. The segmentation mask grows from landmark positions on cusps until some significant change in curvature is found. Such change in curvature should represent the tooth–gingiva boundary. However, in cases where the input mesh lacks high-frequency information, such boundaries are blurred, and the method fails. The performance is also limited in cases where the central pit of the occlusal surface is deep or dental appliances are present. Our proposed solution does not show evidence of these artifacts but may occasionally introduce errors in teeth areas occluded during rendering, such as narrow inter-dental spaces. Figure 6 depicts more results on test samples.

#### 5.4.4. LMVSegRNN Is Robust towards Mesh Tessellation

Naturally occurring small changes in mesh topology and geometry may be observed in the analyzed data, as they are influenced by the scanning properties, which may vary depending on the clinic, scanner, etc. We believe that methods should learn the structural nature of the real-world object represented by the mesh rather than the discrete approximation encoded in raw mesh data. To examine this, we evaluate the robustness to various geometry tessellations to see how individual methods handle changes that do not affect object semantics. The test sets are modified with the subsequent methods:Random vertex displacements (RVD). Vertices are displaced by a value from 〈0.0mm, 0.2mm〉, sampled uniformly. The maximal displacement value is set so the perturbation does not affect shape semantics and is determined according to the typical error of modern intra-oral scanners [27].Planar flipping optimization (PFO). We utilize edge flip operation to locally remesh the input shape, increasing local triangle quality. Again, a similar procedure is applied during training data augmentation.Explicit remeshing (ER). We apply local remeshing operators edge flip, edge collapse, relax, and refine, to obtain isotropically remeshed geometry.

The obtained results are summarized in Table 3. From the results, the following conclusions emerge. The highest drop in performance is observed when RVD transformation is applied, as it introduces the most considerable changes in geometry. All methods are collectively sensitive to vertex displacements, with an average performance drop of approximately 3% on *W-IoU* metric. When visually examined, we find that the measured increase in error is reflected in more jagged region boundaries. The remaining remeshing adjustments aim to increase the mesh quality by introducing changes in the topology. Though, in general, no significant performance drop is observed, the smallest performance drop is measured in the results of our projection-based method.

#### 5.4.5. (Un)Projection Does Not Cause Information Loss

One of the frequent arguments against projection methods is the loss of information when projecting a 3D shape into the 2D feature map space and subsequent information unprojection back to mesh cells. In our proposed method, there is inherently no loss of information, even on unsampled meshes. This is primarily due to local per-teeth processing, a reasonable number of views (49) and camera extrinsic setup, and sufficient resolution of the 2D feature maps (256 × 256). (Un)projection loss of information would be reflected by the generated masks containing holes. Figure 7 shows an example of two teeth in parameterization space obtained by prediction on a mesh with fine geometry: 312,854 faces, where the dataset mean is 203,974 ± 49,800 faces. Occasional unprojection errors are present near the tooth–gingiva boundary. The small errors in this part of the masks are driven by the complex geometry near the gingiva and the lower number of views from which a given part of the surface is visible. However, most cases contain no holes, even on meshes with high-resolution geometric details.

### 5.5. Discussion

To summarize, we demonstrated that a projection-based method that might appear to introduce potential information loss during rendering can precisely segment challenging cases in 3D dental shapes when thoughtfully designed. The results demonstrate that the method outperforms several established point- and edge-based methods [8,9,17], as well as a Graph Cut-based conventional method specifically tailored for this task, indicating its highly competitive performance. The proposed method does not produce errors where, for example, half of the tooth is not segmented due to the low amount of geometric details or scanned dental appliances. This is the case with conventional segmentation methods, and it is considerably time consuming to fix when employed in actual clinical work; our method could decrease this overhead. The results hold for both maxillas and mandibles, as well as among individual tooth types (incisors, canines, premolars, and molars), without any significant variance in performance.

Although our projection-based approach achieves high-quality results, it has certain limitations in handling cases with very narrow inter-dental spaces. Our network is limited in precisely segmenting corresponding cells if the space is extremely narrow due to occlusions.

In the future, our work could be extended in several ways. Recent work [5] has demonstrated the benefit of splitting the geometry and curvature information into two streams and fusing their complementary information to learn more discriminative shape representations. It would be intriguing to see if the same applies to projection-based approaches that employ depth and normal maps, like ours.

Looking forward, further attempts could combine the benefits of projection and geometric methods. Such a combination could lead to a more expressive encoding of the input tooth shapes by analyzing visual and geometric aspects as in [28]. We believe that such fusion could effectively suppress the errors of our method since the inter-dental space could be segmented based on geometric information.

Future research should further develop and confirm the initial findings by comparing the results of our projection-based method with more point-, edge- and graph-based approaches.

Our framework can also be applied to other areas of digital dentistry. For instance, a method inspired by our approach could be used to automate tasks such as margin line detection or tooth preparation segmentation in automatic crown design [29].

### 5.6. Contributions

Lastly, we briefly summarize the contributions:Streamlined segmentation. Our method operates in a detect-and-segment manner, where the detected surface landmarks (rather than tooth centroids) offer practical advantages for clinicians, enabling quick corrections compared to the laborious adjustments of segmented 3D regions. This demonstrates significant potential to reduce human efforts in orthodontic treatment planning.Information-rich projections. Alongside the local tooth processing, we show that the rendered maps capture the 3D shape with minimal information loss.Highly competitive performance. We achieve an average weighted IoU score of 0.97122±0.038 and a Hausdorff distance at the 95th percentile of 0.49012±0.571 mm. We provide a comparative analysis with other segmentation approaches.Release of Poseidon3D dataset. A new challenging dataset of real-world orthodontic cases (teeth crowding, damaged and missing teeth, and teeth with visible roots) is introduced: Poseidon’s Teeth 3D (Poseidon3D).

## 6. Conclusions

We have introduced a novel projection-based framework specifically tailored to the task of teeth segmentation in 3D surface dental scans. To minimize the information loss, we defined the task to generate binary segmentation masks in a detect-and-segment manner, introducing local tooth analysis, followed by custom post-processing derived from multi-view certainty. We have demonstrated that a 2D approach is able to not only produce promising results for practical applications but it can also have a competing performance, which we have shown by comparing our method with several established geometric approaches. We also release the complete dataset publicly to elevate further research in this field.

## Figures and Tables

**Figure 1 bioengineering-11-01014-f001:**
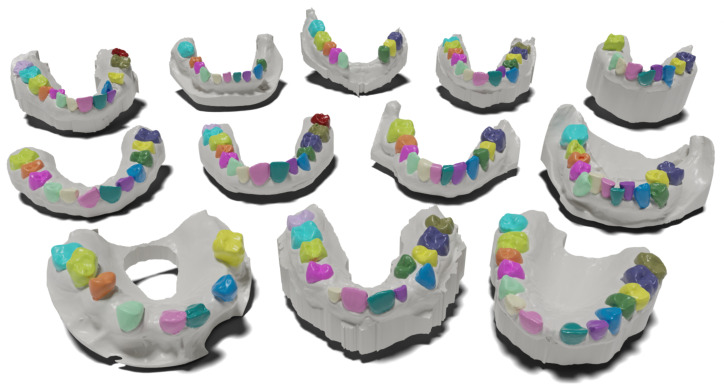
Representative cases from Poseidon3D, a new public dataset of anatomically complex 3D dental scans introduced in this work. Focus on the anatomical variability across individual subjects. Colored teeth regions are the predicted segmentation masks by the proposed projection-based method. Zoom in to see the intricate anatomical details.

**Figure 2 bioengineering-11-01014-f002:**
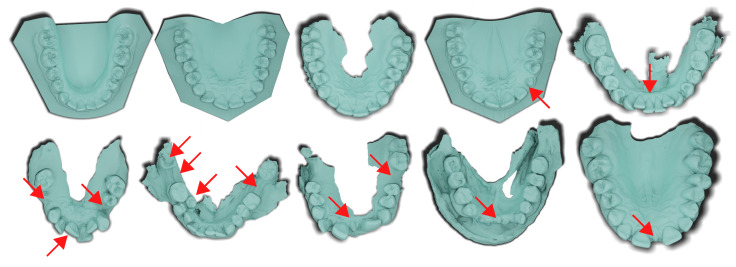
Randomly sampled cases from *3DTeethSeg22* [18] dataset (**upper row**) and Poseidon3D, the dataset presented in this work (**lower row**). Focus your attention on the highlighted regions marked by arrows in the figure. These areas represent natural anatomical challenges (teeth crowding, missing, damaged teeth, retention, diastema, and others) relentlessly present in orthodontic cases. In 3DTeethSeg22, such features also appear but not as commonly as in our dataset.

**Figure 3 bioengineering-11-01014-f003:**
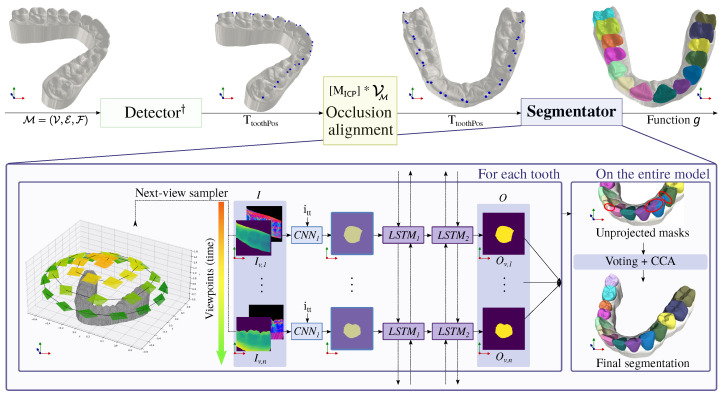
Outline of our proposed projection-based tooth segmentation method. ^†^ As the detector, we employ the multi-view landmark detection method presented in [16]. We utilize the detected teeth position information for transforming the input mesh to the canonical form using the ICP algorithm (occlusion alignment), localizing the segmentation over individual teeth (next-view sampler), and conditioning the training and inference via input tooth type signal ($i_{tt}$ scalar). Generated 2D segmentation maps from *n* viewpoints are correlated by recurrent network, unprojected to mesh cells, and post-processed to suppress any region ambiguities (region voting) and isolated triangles (CCA).

**Figure 4 bioengineering-11-01014-f004:**
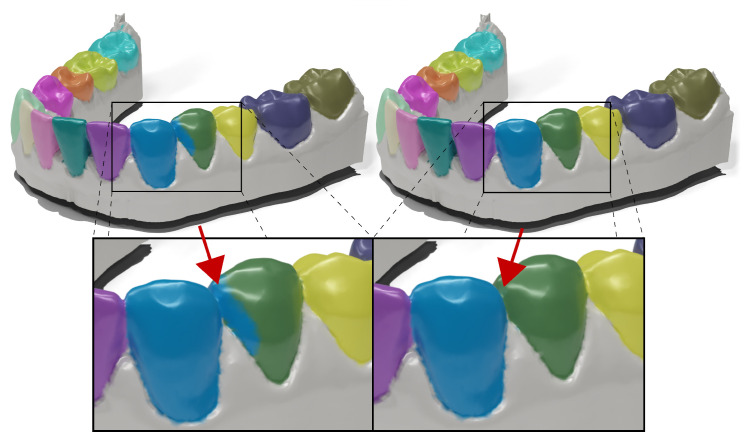
Qualitative assessment of the method without (**left**) and with (**right**) the region voting. In the former, predictions are back-projected to mesh sequentially in canonical order, for example, from R8 to L8. Blue and green region cells with misclassified labels are effectively reassigned based on a higher certainty value.

**Figure 5 bioengineering-11-01014-f005:**
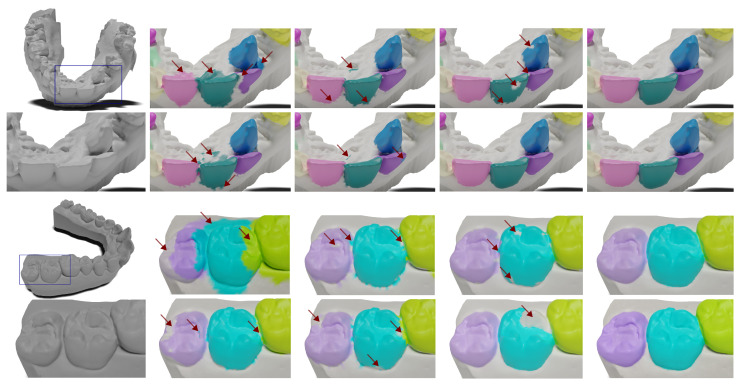
Visualization of two segmentation results generated by the competing methods, our proposed method, and ground truth. From left to right, top row first: input mesh, PointNet without normal vectors, PointNet with normal vectors, SparseMeshCNN with attention blocks, our LMVSegRNN, zoomed-in input, PointNet++ without normal vectors, PointNet++ with normals, Graph Cut, ground truth. Zoom in to better see the details.

**Figure 6 bioengineering-11-01014-f006:**
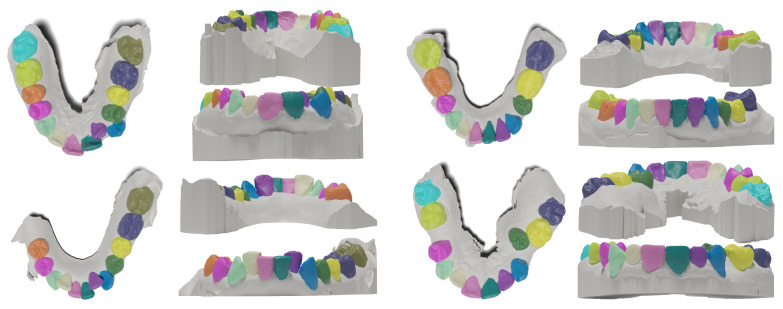
Detailed visualization of the results of the best-performing setup of LMVSegRNN on four randomly selected complex test cases. For each sample, we visualize the results from the occlusal view, and the lingual and buccal surfaces in orthographic projections. Zoom in for details.

**Figure 7 bioengineering-11-01014-f007:**
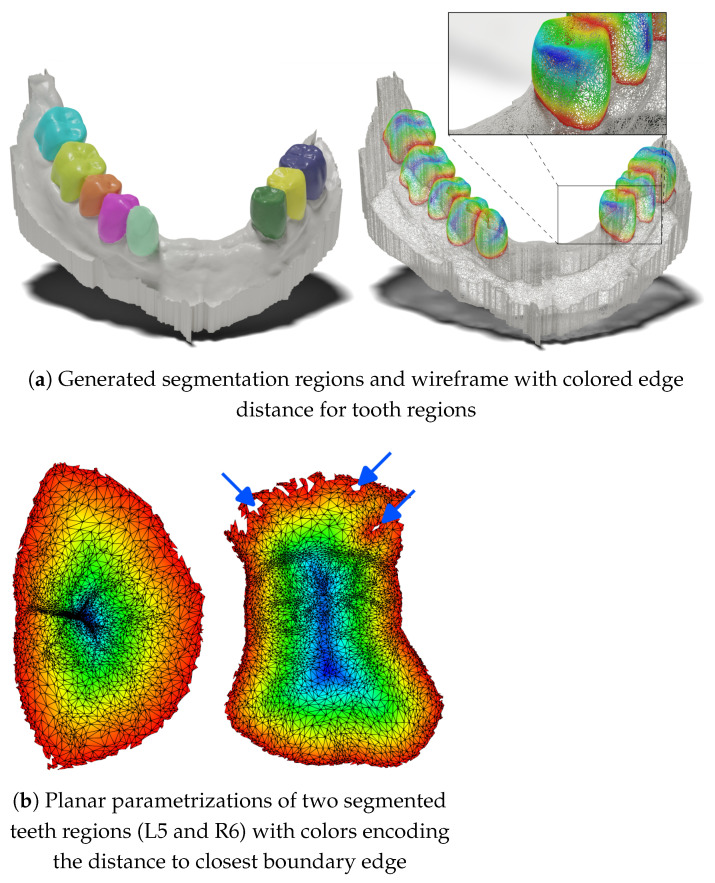
Demonstration information preservation during the (un)projection process. (**a**) Visualization of the segmentation results of one high-resolution test case. (**b**) Two generated masks from the case in parametrization space (unwrapped to 2D). In most cases, the generated teeth regions are consistent without any holes. Occasionally, masks contain small projection errors near the tooth–gingiva boundary, see blue arrows.

**Table 1 bioengineering-11-01014-t001:** Ablation studies on various method components. Ds: depth information at the input in the form of depth maps. Ns: curvature information at the input in the form of normal maps. $i_{tt}$: tooth type input signal. Deep s.: deep supervision. LSTMs: bi-directional ConvLSTM units for inter-viewpoint content. Voting: region voting for correcting unprojected segmentation masks. CCA: connected component analysis for removing isolated region cells. Results in bold are for the best-performing method.

							Basic Cases	Complex Cases
* Ds *	* Ns *	$i_{\mathrm{tt}}$	* Deep s. *	* LSTMs *	* CCA *	* Voting *	W-IoU↑	W-IoU↑
✓							0.91759 ±0.087	0.91857 ±0.067
	✓						0.90898 ±0.093	0.88943 ±0.101
✓	✓						0.92045 ±0.125	0.92670 ±0.011
✓	✓	✓					0.92811 ±0.103	0.92564 ±0.065
✓	✓	✓	✓				0.92826 ±0.090	0.92980 ±0.098
✓	✓	✓	✓	✓			0.94796 ±0.105	0.94438 ±0.051
✓	✓	✓	✓	✓	✓		0.95145 ±0.105	0.95132 ±0.047
✓	✓	✓	✓	✓	✓	✓	0.97676 ±0.084	0.97122 ±0.038

**Table 2 bioengineering-11-01014-t002:** Comparison of semantic tooth segmentation between LMVSegRNN and existing methods using *Poseidon3D* dataset. By *n*, we refer to normal vectors as a part of input features. By *ab* (attention blocks), we refer to incorporating edge self-attention blocks in the two bottom stages of SparseMeshCNN. Results in bold are for the best-performing method.

Seg. Method	Basic Cases		Complex Cases	
	W-IoU↑	$d_{H95}↓$	W-IoU↑	$d_{H95}↓$
PointNet (*w/o n*)	0.822 ±0.065	1.959 ±1.611	0.785 ±0.090	2.437 ±1.231
PointNet (*w. n*)	0.928 ±0.021	1.120 ±1.773	0.921 ±0.039	1.309 ±1.262
PointNet++ (*w/o n*)	0.914 ±0.043	0.983 ±1.522	0.907 ±0.058	1.396 ±1.363
PointNet++ (*w. n*)	0.943 ±0.053	1.033 ±1.039	0.935 ±0.051	1.022 ±1.129
SpMeshCNN (*w/o ab*)	0.923 ±0.095	0.784 ±1.225	0.920 ±0.092	0.893 ±1.103
SpMeshCNN (*w. ab*)	0.948 ±0.063	0.623 ±0.794	0.918 ±0.057	0.858 ±1.074
Graph Cut	0.961 ±0.184	1.013 ±2.816	0.945 ±0.109	0.985 ±1.794
LMVSegRNN	0.976 ±0.084	0.458 ±0.993	0.971 ±0.038	0.490 ±0.571

**Table 3 bioengineering-11-01014-t003:** Robustness of methods towards various geometry tessellations. Abbreviations of topology and geometry adjustment methods: RVD: Random Vertex Displacement, PFO: Planar Flipping Optimization, ER: Explicit Remeshing. Blue text represents the performance change compared to the reference performance (without any topology or geometry changes) of given method. Presented point-based methods contain both 3D coordinates and normal vectors of mesh cells at the input. SparseMeshCNN contains edge self-attention blocks in two bottom stages of the encoder. Bold text represent the best-performing setup.

Method	Topology Adjustment & Performance			
	Reference	RVD	PFO	ER
	W-IoU↑	W-IoU↑	W-IoU↑	W-IoU↑
PointNet	0.921 ±0.039	(−2.4%) 0.897 ±0.044	(−1.1%) 0.910 ±0.073	(−0.7%) 0.914 ±0.060
PointNet++	0.935 ±0.051	(−3.5%) 0.900 ±0.058	(−1.3%) 0.921 ±0.098	(−0.8%) 0.927 ±0.049
SpMeshCNN	0.918 ±0.057	(−2.8%) 0.890 ±0.065	(−1.8%) 0.900 ±0.052	**(+0.2%)** 0.920 ±0.082
Graph Cut	0.954 ±0.109	(−3.5%) 0.918 ±0.133	(−1.6%) 0.937 ±0.101	(−1.3%) 0.940 ±0.113
LMVSegRNN	0.971 ±0.038	**(−1.8%)** 0.953 ±0.086	**(−0.3%)** 0.968 ±0.055	(−0.5%) 0.966 ±0.092

## Data Availability

The original data presented in the study are openly available at https://tiborkubik.github.io/lmvseg-poseidon3d/ (accessed on 10 October 2024).

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
