# Peer review of "LMVSegRNN and Poseidon3D: Addressing Challenging Teeth Segmentation Cases in 3D Dental Surface Orthodontic Scans"

_bioengineering, 2024, doi:10.3390/bioengineering11101014_

Round 1
Reviewer 1 Report
Comments and Suggestions for Authors
This research article provides a new technology to overcome the 3D printing and project methods related to tooth alignments in orthodontic consideration. The authors offer a projection-based method of tooth segmentation in 3D surface dental scans. Congratulations to the authors who developed and generated datasets for the public to use in further research. The novelty is good. I think this article may be ready to be published.
Author Response
Thank you, we appreciate your congratulations.
Reviewer 2 Report
Comments and Suggestions for Authors
Digitization of dental surgery is the future trend of dentistry. And this situation continues to evolve. This study introduces a novel projection-based framework specifically targeted at the task of tooth segmentation in 3D surface dental scanning. Research shows that this method achieves satisfactory results. The structure of this manuscript is quite complete. I have no objections to this manuscript. But there are still suggestions. The advantages and disadvantages of using this digital technology in actual clinical dental work compared to traditional laboratory tooth segmentation methods can be illustrated in the discussion. In addition, when this digital technology is applied to actual dental clinical work, what is the expected effect? What other applications does this digital technology have in dentistry?
Author Response
Comment 1: The advantages and disadvantages of using this digitisation technique in actual clinical dental work as compared to conventional laboratory tooth segmentation methods can be described in the discussion.
Response 1: Thank you for this valuable comment. We agree with this point and we added a new sentence to the discussion that explicitly describes the advantages of this segmentation technique: The proposed method does not produce errors where, for example, half of the tooth is not segmented due to the low amount of geometric details or scanned dental appliances. This is the case with conventional segmentation methods, which is considerably time-consuming to fix when employed in actual clinical work, and our method could decrease this overhead. The disadvantages are already discussed further in the discussion, within the future work text.
Comment 2: Also, what are the expected results when this digitisation technique is applied to actual clinical dental work?
Response 2: The expected results are now clear in the discussion, thanks to the added text for the previous comment. It should decrease the time clinicians spend on fixing incorrect segmentation results of conventional approaches.
Comment 3: What are the other applications of this digitisation technique in the field of dentistry?
Response 3: Thank you for this comment, it is really important to point out the application of the proposed method in other branches of digital dentistry. For that reason, we've added following text to the discussion: Our framework can also be applied to other areas of digital dentistry. For instance, a method inspired by our approach could be used to automate tasks such as margin line detection or tooth preparation segmentation in automatic crown design~\cite{Zhang2019TheEM}. We believe that the method could be simply adapted for this task in digital crown design and restoration. We also added a citation for another work that proposes a framework for this task
Reviewer 3 Report
Comments and Suggestions for Authors
The manuscript seems interesting and unique. The authors should address the following issues to improve its quality as following:
- Authors should use passive voice in scientific writing.
- Authors should avoid advertising approach in writing and discuss the details in scientific way.
- Objects in some figures are too small to see. Authors should enlarge them or split them into multiple figures.
- Discussion is too short. Although the approach used is innovative and new to scientific community, authors should point out the possible limitations and therefore recommend future development.
- Conclusion section can be summarized in bullet point.
Author Response
We would like to thank the reviewer for their valuable comments. Please find our responses below.
Comment 1: Authors should use passive voice in scientific writing.
Response 1: We adjusted the text to passive voice in several parts of the text, primarily in chapters 2, 3, and 4, that present the dataset, methodology and implementation details.
In total, we changed 17 sentences from active to passive voice.
We left an active voice in contributions and in parts of the text where we present our observations about the results or decisions.
Comment 2: Authors should avoid advertising approach in writing and discuss the details in scientific way.
Response 2: Of course, advertising the method by providing strong quantitative and qualitative results is crucial in academic writing. However, we believe that some aspects of the approach are better described in words. For example, in terms of measured metrics, the achieved results for conventional approach is not that far from the results of our method. However, the character of the error is really different, and causes more edits and tweaks to fix the generated masks. This is critical for the streamlined workflow, and we found advertising this in writing is more clear, especially when coupled with strong quantitative analysis that we provide.
Comment 3: Objects in some figures are too small to see. Authors should enlarge them or split them into multiple figures.
Response 3: Regarding this comment, the most critical figure, in our eyes, is Figure 6. Our first intention was to have the subfigures side-by-side, but we agree that it decreased the amount of details visible. So, we enlarged both subfigures and put them under each other. We also added a note to the description of Figure 4, that the more details are visible when zoomed in.
Comment 4: Discussion is too short. Although the approach used is innovative and new to scientific community, authors should point out the possible limitations and therefore recommend future development.
Response 4: We made the discussion longer since we also added text based on the suggestions from other reviewers. It now contains more clear statements of the advantages and disadvantages of our method compared to conventional techniques. It now also describes how our method could be applied to other tasks in other branches of digital dentistry, and we added a brief summary of contributions at the end of the discussion.
Reviewer 4 Report
Comments and Suggestions for Authors
The topic is very interesting, but some concerns are raised especially regarding the manuscript form. There is no aim of the study. The subsection Contributions should be incorporated in the Discussion. The sample size analysis is missing. The background literature review should be expanded.
Author Response
We would like to genuinely thank the reviewer for their comments.
Comment 1: There is no aim of the study.
Response 1: We really missed the detail about the aim of the study in the manuscript. Therefore, we explicitly mention the aim in the introduction by adding following sentence:
...An automatic solution capable of segmenting teeth in 3D surface scans robustly to all mentioned aspects is indispensable to streamlining clinicians' workflow and focusing more attention and focus on treatment planning. The aim of this study is to provide a public dataset of such cases and also a method that handles the segmentation of these cases with the minimum necessary adjustments. ...
Comment 2: The subsection Contributions should be incorporated in the Discussion.
Response 2: We added a brief summary of our contributions as a subsection in Discussion.
Comment 3: The sample size analysis is missing.
Response 3: In the section 2: Poseidon3D: Dataset of Anatomically Complex Dental Shapes, we describe the sample size of the dataset in terms of the total number of cases, as well as how this number divides into maxillas and mandibles. We also compare this with regard to another public dataset. Other information could not be provided, since the data are patient data and are anonymized.
Comment 4: The background literature review should be expanded.
Response 4: In our literature overview, we mainly focused on the technical papers from machine learning and 3D shape analysis, and for the task-specific methods for mesh teeth segmentation. So, based on the comment, we added two literature sources from the clinical publications, to support the paper on the importance of tooth segmentation in clinical procedures, particularly in orthodontics. They are referred in introduction [1] and [2]. We also added one extra source in the discussion to show to which other branches of digital dentistry our framework could be extended.
Round 2
Reviewer 2 Report
Comments and Suggestions for Authors
Digitization of dental surgery is the future trend of dentistry. And this situation continues to evolve. This study introduces a novel projection-based framework specifically targeted at the task of tooth segmentation in 3D surface dental scanning. Research shows that this method achieves satisfactory results. The structure of this manuscript is quite complete. I have no objections to this manuscript. And regarding the previous suggestions, the authors have completed appropriate revision. I am very grateful to the authors for your/their efforts on this article. The overall recommendation is to accept this manuscript in present form.
Reviewer 4 Report
Comments and Suggestions for Authors
The changes are not highlighted in the revised version of the manuscript, please do it.